# Fucoidan-Incorporated Composite Scaffold Stimulates Osteogenic Differentiation of Mesenchymal Stem Cells for Bone Tissue Engineering

**DOI:** 10.3390/md20100589

**Published:** 2022-09-21

**Authors:** Yashaswini Devi G.V., Apoorva H Nagendra, Sudheer Shenoy P., Kaushik Chatterjee, Jayachandran Venkatesan

**Affiliations:** 1Biomaterial Research Laboratory, Yenepoya Research Centre, Yenepoya (Deemed to be University), Mangalore 575018, India; 2Stem Cells and Regenerative Medicine and Yenepoya Research Centre, Yenepoya (Deemed to be University), Mangalore 575018, India; 3Departmental of Materials Engineering, Indian Institute of Science, Bangalore 560012, India

**Keywords:** bone regeneration, fucoidan, graphene oxide, hydroxyapatite, sodium alginate

## Abstract

Globally, millions of bone graft procedures are being performed by clinicians annually to treat the rising prevalence of bone defects. Here, the study designed a fucoidan from *Sargassum ilicifolium* incorporated in an osteo-inductive scaffold comprising calcium crosslinked sodium alginate-nano hydroxyapatite-nano graphene oxide (Alg-HA-GO-F), which tends to serve as a bone graft substitute. The physiochemical characterization that includes FT-IR, XRD, and TGA confirms the structural integration between the materials. The SEM and AFM reveal highly suitable surface properties, such as porosity and nanoscale roughness. The incorporation of GO enhanced the mechanical strength of the Alg-HA-GO-F. The findings demonstrate the slower degradation and improved protein adsorption in the fucoidan-loaded scaffolds. The slow and sustained release of fucoidan in PBS for 120 h provides the developed system with an added advantage. The apatite formation ability of Alg-HA-GO-F in the SBF solution predicts the scaffold’s osteointegration and bone-bonding capability. In vitro studies using C3H10T1/2 revealed a 1.5X times greater cell proliferation in the fucoidan-loaded scaffold than in the control. Further, the results determined the augmented alkaline phosphatase and mineralization activity. The physical, structural, and enriching osteogenic potential results of Alg-HA-GO-F indicate that it can be a potential bone graft substitute for orthopedic applications.

## 1. Introduction

Bone defects and diseases significantly reduce the quality of human life. Bone tissue naturally possesses a good healing ability. Still, if the defect site is greater than the critical size threshold (typically > 2 cm), it fails to heal on its own and needs an artificial intervention [1,2]. Over 15 million bone fracture cases and around 2 million bone graft procedures are recorded per annum [3,4]. Current treatment for bone defects includes the utilization of autologous and allogeneic tissue grafts. However, they are associated with complications that include limited graft availability, donor site morbidity, and a loss of osteoconductive and osteo-inductive ability in sterilization [4,5]. Synthetic bone grafts emerged as a promising alternative to address autograft and allograft limitations. The implanted scaffolds comprised degradable polymers with bone repairability. The ideal functional scaffold material in bone tissue engineering should have specific characteristics, including a biocompatible, biodegradable, porous surface for cell growth and mechanical support that meet the implanted tissue [6]. The factors that affect these properties are structure and material composition [7]. Hence, bone regenerative research in the past few decades has encountered various natural–synthetic polymers and ceramics composites.

Alginate is a widely used natural biomaterial in tissue engineering due to its biocompatible and biodegradable properties [1]. Its ability to become readily crosslinked with the divalent calcium ions and its structural similarity with the extracellular matrix makes it highly utilized in biomedical areas [2]. In addition, alginate scaffolds lack a cell adhesivity property, which is a significant material drawback. Thus, to overcome this limitation, alginate is either modified with cell adhesion peptides or combined with other biomaterials with higher cell adhesion sites [3,4]. Nano-hydroxyapatite (HA) is an inorganic material widely used for bone tissue engineering due to the biomimicking properties of natural bone. HA is a stable calcium phosphate at room temperature, with a 1.67 molar ratio of Ca/P. By blending with alginate, HA has gained significant attention in tissue engineering applications. The composite of Alg-HA is one of the best-studied biomaterials for bone tissue engineering applications. Studies have scrutinized the impact of nano-hydroxyapatite as an excellent antibacterial agent with improved mechanical properties [5]. Furthermore, researchers have elucidated the Alg-HA scaffolds with the potential to mineralize and differentiate stem cells [6]. In addition, Alg-HA has a beneficiary effect as a bone substitute or filler. Various studies have used Alg-HA biocomposites as a bone substitute, validated by the in vivo reports supporting femur bones’ formation and new bone formation in the calvaria defect rat model [7,8]. However, HA’s slow degradation and brittleness limit its use in bone tissue engineering applications [9,10].

Carbon-based nanomaterials are gaining remarkable attention in biomedical applications due to their unique properties, including a large surface area and high mechanical potency [11]. Carbon-based materials have been extensively studied for drug delivery, tissue engineering, and biosensor applications [12,13]. Graphene oxides (GO) are oxidized forms of graphene with a single layer of active oxygen-containing functional groups such as hydroxyls, epoxides, and peripheral carboxylic groups [14]. Recent studies emphasize that a composite with GO integration significantly enhances mechanical strength and supports osteoblast adhesion and bone regeneration [15]. On the other hand, the GO-HA composite has achieved an increased interest in orthopedic applications [16].

Fucoidan is a sulfated polysaccharide commonly observed in the cell wall of brown algae. Fucoidan comprises fucose, glucose, galactose, mannose, xylose, rhamnose, and uronic acid [17,18]. Several studies have proven the spectra of bioactivities of fucoidan, including antioxidant, anti-inflammatory, antiviral, and anti-tumor, and its role in bone repair [17,18,19,20,21]. Furthermore, these bioactivities are tailorable with the advanced extraction process and the post-processing steps of fucoidan. For example, the enzymatic degradation technique was used to create low and medium-molecular-weight fucoidan with altered bioactivities to explicate the structure–function relationships [22]. In addition, reports reveal that fucoidan can significantly enhance bone cell activity and induce osteogenic differentiation in mesenchymal stem cells [23]. However, the bioactivity of fucoidan is based on its structural features, which depend on environmental parameters, seasons, and extraction techniques [24,25].

To mimic the natural function of bone in terms of the porosity, osteoinduction, surface topography, and mechanical strength of the scaffold, we developed novel fucoidan-loaded Alg-HA-GO composite scaffolds for bone tissue regeneration. The scaffolds were developed by forming a calcium-crosslinked porous network of polymers via the freeze-drying method. The developed composites were examined in surface morphology, degradation, swelling behavior, and mechanical properties.

## 2. Results

### 2.1. General Observation

Figure 1 represents a schematic diagram of the composite scaffold preparation. The designed calcium crosslinked Alg-HA, Alg-HA-GO, and Alg-HA-GO-F scaffolds were fabricated via the freeze-drying technique. The developed matrices are rigid with a highly porous structure. Initially prepared, the 3% Alg solution was yellow, followed by the slow addition of HA crystals to avoid clump formation. After the HA addition, the solution turned into a white gel. Next, the addition of GO resulted in the final black color of the synthesized scaffolds. Further, the fucoidan was slowly loaded.

### 2.2. FT-IR Analysis

FT-IR analysis was performed on the developed composite scaffolds of Alg-HA, Alg-HA-GO, and Alg-HA-GO-F (Figure 2A). The FT-IR spectra of HA, Alg, GO, and F from *Sargassum ilicifolium* are shown in Figure 2B.

The FT-IR spectra of HA showed sharp bands at 1019 cm^−1^ and 961 cm^−1^ assigned to the symmetric and antisymmetric stretching vibrations of P-O bonds present in the PO_4_^2−^ group. The bands at 599 cm^−1^ and 562 cm^−1^ correspond to the antisymmetric bending of the PO_4_^2−^. The symmetric bending vibrations of the PO_4_^2−^ group were attributed to the bands around 467 cm^−1^. The characteristics peaks at 1455 cm^−1^ and 1416 cm^−1^ was for the CO_3_^2−^ group [26,27,28]. The spectrum of unmodified sodium alginate showed strong peaks of the hydroxyl group at 3340 cm^−1^. The peaks at 1605 cm^−1^ and 1432 cm^−1^ was assigned to the carboxyl group’s symmetric and asymmetric stretching vibrations. The bands between 1079 cm^−1^ and 1012 cm^−1^ was assigned to its glucuronic (G) and mannuronic (M) acid units. The β-glycosidic linkage between G and M is found in 815 cm^−1^ signals [29,30]. The spectrum of nano graphene oxide powder indicated no major functional groups [31,32]. Fucoidan from *Sargassum ilicifolium* revealed the characteristics functional group of polysaccharides. The absorption peak at 3369 cm^−1^ corresponds to the stretching vibration of the hydroxyl group. The band at 1418 cm^−1^ and 1623 cm^−1^ was assigned to O-H bending and stretching of an alkene conjugated to the COOH group, respectively [33]. The typical absorption signal at 1219 cm^−1^ represents S=O stretching vibrations [20,33]. A strong band at 1028 cm^−1^ represents C-O stretching vibration related to a C-O-SO_3_ group [34]. The peak at 821 cm^−1^ corresponds to the C-O-S stretching vibration of sulfate ester located at the C-4 axial position of fucose units [17,33].

The FT-IR spectra of Alg-HA, Alg-HA-GO, and Alg-HA-GO-F are shown in Figure 2(Aa)–(Ac)). The FTIR spectra of calcium crosslinked composite Alg-HA showed characteristic peaks of sodium alginate at 1601 cm^−1^ and 1435 cm^−1^ (asymmetric and symmetric vibrations of -COO-). They are attributed to the ionic interaction between the carboxylate anion of Alg and the calcium cation of HA [5]. The bands at 3341 cm^−1^ and 1078 cm^−1^ contribute to the O-H and C-O stretch, respectively. The peaks detected at 599 cm^−1^, 565 cm^−1^_,_ and 939 cm^−1^ were assigned to the bands of the PO_4_^3−^ group, indicating the existence of the HA phase [35,36]. Adding 0.1% of GO in Alg-HA significantly shifted the FT-IR peaks (Figure 2(Ab)). A slight shift from 3341 cm^−1^ to 3368 cm^−1^ observed in the Alg-HA-GO spectra indicates the materials’ interaction. The interaction may be associated with the hydrogen bonding between GO and Alg. In addition, peak shifts from 1601 to 1612 cm^−1^, 1078 to 1086 cm^−1^, and 1008 to 1024 cm^−1^ were even observed. This might be because of the crosslinking interaction of GO with Ca^2+^ ions [37,38]. The FT-IR spectrum of Alg-HA-GO-F is shown in Figure 2(Ac). The addition of fucoidan exhibits a characteristic band at 822 cm^−1^ for the sulfate group. The stretching frequency of 3360 cm^−1^, 1604 cm^−1^, 1429 cm^−1^, and 1026 cm^−1^ may come from the Alg [39,40].

### 2.3. X-ray Diffraction Analysis of the Scaffolds

The X-ray diffraction spectra of Alg-HA, Alg-HA-GO, and Alg-HA-GO-F are shown in Figure 2C. In all of the XRD spectra, the diffraction peaks at 25.65, 28.80, 31.58, 32.7, 34.1, 39.54, 46.47, and 49.23 correspond to the indices (0 0 2), (2 1 0), (2 1 1), (3 0 0), (2 0 2), (3 1 0), (2 2 2), and (2 1 3), which correspond to HA. The phase peaks of all three composites comply with the standard HA [27,36,41]. The recorded spectra of Alg-HA, Alg-HA-GO, and Alg-HA-GO-F were close to HA. Using the higher amount of HA and Alg, and as the Alg is amorphous, major peaks contributed to HA.

### 2.4. Thermogravimetric Analysis

Thermogravimetric analysis in each synthesis step investigated the mass loss at different temperatures. Figure 2D demonstrates the TGA curves of the Alg-HA, Alg-HA-GO, and Alg-HA-GO-F composite scaffold. The Alg-HA composite showed an initial weight loss of almost 2.8% between ambient temperature and 240 °C, primarily attributed to the evaporation of water molecules. The second weight loss was at the temperature range of 240 °C to 540 °C, where the primary degradation of alginate due to decarboxylation happens at 240 °C to 300 °C [42]. The recorded initial weight loss in the thermogram of the Alg-HA-GO composite scaffold can be due to the removal of trapped water molecules. The second mass reduction from the temperature range between 257 °C and 454 °C is attributed to the pyrolysis of oxygen-containing functional groups such as carboxyl, hydroxyl, and epoxides in the form of carbon monoxide and carbon dioxide [14]. For the Alg-HA-GO-F composites scaffold, adding fucoidan increased the decomposition rate in the temperature between 272 °C and 639 °C, corresponding to the devolatilization and decomposition of the fucoidan [43], and the degradation of other components used in the scaffold designing. Each composite observed distinct thermal behaviors due to the interactions leading to the change in molecular properties, specifically by its crosslinking.

### 2.5. Surface Analysis of the Synthesized Scaffolds

Lyophilization procedures majorly influence the surface morphology of the scaffolds, which leads to porous structures in the composite scaffolds. The freeze-drying technology utilizes the refrigerant that leads to ice crystal formation, which is transferred into the pore via the sublimation process, leaving behind the highly interconnected architecture [33]. Further, the freeze-dried scaffolds were characterized for their surface morphology and elemental analysis was conducted using optical microscopy, FESEM, and EDAX.

The FESEM images of cross-sections of scaffolds show a highly interconnected pore distribution as shown in Figure 3A. The Image J software analyzed the relative pore size of the composites. The analysis revealed that the pore size of the Alg-HA, Alg-HA-GO, and Alg-HA-GO-F scaffolds are 96–250 μm, 100–320 μm, and 108–460 μm, respectively. The average pore size of scaffolds was increased with the addition of graphene oxide and fucoidan. The obtained pore size range agrees with previous reports [39,40,41,42,43,44]. Figure 3B displays optical microscopy images of the synthesized scaffolds, disclosing its porous surface with the uniform distribution of components within the scaffolding network. Figure 3C illustrates a qualitative and quantitative elemental compositional analysis of EDAX. The spectra of Alg-HA and Alg-HA-GO were found to have carbon, oxygen, calcium, phosphorous, and chlorine. In addition, the EDAX spectra of Alg-HA-GO-F disclosed the presence of sulfate from sulfated polysaccharide or fucoidan, along with other elements. The measured calcium-to-phosphate molar ratio of Alg-HA, Alg-HA-GO, and Alg-HA-GO-F was 1.5, 1.8, and 2.2. This owes to the addition of negatively charged graphene oxide and fucoidan resulting in an increased Ca^2+^ and the crosslinking agent content being directly proportional to the calcium-to-phosphate ratio.

### 2.6. Atomic Force Microscopy

The fabricated three types of scaffolds showed different surface roughness (Figure 4). AFM assessment for the roughness displayed that the HA with SA was seemingly rough with crack structures. In comparison, the addition of GO and fucoidan showed a decrease in the roughness profile (Figure 4D). Hence, the composite Alg-HA-GO-F showed fewer scars with a uniform structure when compared to the Alg-HA-GO and Alg-HA. The obtained R_a_ values were 426.8 nm for Alg-HA with a higher surface roughness, 342.9 nm for Alg-HA-GO with medium roughness, and 259.1 nm for Alg-HA-GO-F with low roughness.

### 2.7. Porosity Measurement

Porosity is a fundamental property that ensures cellular migration to support vascularization during tissue formation [44]. Here, the porosity of the developed freeze-dried scaffolds was evaluated by the liquid displacement method and is shown in Figure 5A. The porosity of the Alg-HA, Alg-HA-GO, and Alg-HA-GO-F of the scaffold was 82.7%, 83.13%, and 87.3%, respectively. The addition of GO and fucoidan observed slight differences. Furthermore, the porosity was increased by adding fucoidan to the composite scaffold [45]. A porosity of greater than 80% is considered an optimum bone tissue engineering scaffold for nutrient supplements [35,46,47].

### 2.8. Swelling Behavior and Water Uptake and Retention Ability of the Scaffolds

The swelling behavior of the scaffolds is an essential parameter in bone tissue engineering. The swelling behavior of the scaffolds was measured through the water uptake and retention ability in a PBS buffer. The results are presented in Figure 5B,C. During the swelling of the scaffolds, the porosity and pore size increases, supporting cell growth and other cellular activities [45]. A maximum swelling behavior was observed in Alg-HA-GO compared to Alg-HA and Alg-HA-GO-F composite scaffolds. The addition of GO increased the hydrogen bonding between the materials. However, the addition of fucoidan decreased the swelling behavior of the scaffold. In addition, the heterogeneity of the sulfated polysaccharide augments the crosslinking constraining the polymer network, which hampers the water uptake, resulting in decreased swelling.

### 2.9. Biodegradation of Scaffolds

Biodegradation is a necessary feature for bone scaffolds, which play a critical role in the bone tissue regeneration process. The biodegradation of scaffolds happens simultaneously with the new bone tissue formation [48,49]. Figure 5D represents the biodegradation of the developed composites. The biodegradation of scaffolds was performed using lysozyme and PBS for 7 days, 15 days, and 30 days. After 30 days for Alg-HA, Alg-HA-GO, and Alg-HA-GO-F, the obtained degradation percentage was 60%, 38%, and 26.7%, respectively. Alg-HA scaffolds degraded faster than Alg-HA-GO and Alg-HA-GO-F scaffolds. Notably, the weight loss is because of the hydrophilic nature of the Alg with its water uptake ability [8]. However, the addition of graphene oxide and fucoidan significantly decreases biodegradation. In addition, an increased crosslinking among the materials Alg-HA, GO, and F contributes to reducing degradation. Thus, the achieved slow degradation will benefit bone tissue engineering applications. Purohit et al. fabricated the graphene oxide and nanohydroxyapatite-reinforced gelatin–alginate nanocomposite. A similar degradation pattern was observed with 53.8, 46.0, 30.45, and 29.4 degradation percentages for gelatin–alginate, nanohydroxyapatite–gelatin–alginate, graphene-oxide–gelatin–alginate, and nanohydroxyapatite–graphene-oxide–gelatin–alginate scaffolds [44]. The weight loss might also be because a highly porous material with a larger surface area furnishes more active sites for the lysozyme reaction [50].

### 2.10. Fucoidan Loading and Release Study

The fucoidan loading capacity and release profile were determined by measuring the absorbance at 260 nm in UV spectroscopy. The resulting loading capacity of the Alg-HA-GO-F scaffold was 98%. The fucoidan release profile was observed in PBS. The initial burst release of fucoidan within 24 h in PBS was carried out for 120 h, as depicted in Figure 6A. Although freeze-drying provides a suitable porous morphology for the release, the polymer degradation rate also plays a crucial role. The scaffolding component alginate degrades faster; as a result, the fucoidan detaches from the polymer matrix and gets released. The initial burst release is attributed to the fraction of fucoidan, which is weakly bound to the polymer matrix rather than encapsulated. The attained constant release rate from the composite affirms that it can be a potential candidate for the drug release study and bone regeneration. The release studies of fucoidan from matrix composites are scanty, whereas researchers have greatly explored the release from fucoidan-loaded nanoparticles. Etman et al. studied the bioactive targeted fucoidan–lactoferrin nanoparticles. The tested fucoidan release from fucoidan–lactoferrin nanoparticles at different pH values (5.5 and 7.4) reports a sustained release for 48 h [51]. Another work on chitosan–fucoidan nanoparticles, without and with a glutaraldehyde crosslink, exhibits, respectively, a 65.5% and 60.6% release of fucoidan within the first 24 h [48]. Recently, a study investigated the thermosensitive delivery system for fucoidan composed of collagen, chitosan, and β-glycerophosphate. The study reports that 60% was released in two days and 80% in the next six days [49]. Thus, the crosslinked multi-material scaffold-loaded fucoidan served to deliver an extended release.

### 2.11. Mechanical Strength of the Scaffolds

Mechanical strength is an essential quality of the scaffold fabricated for bone tissue engineering. The load-bearing capacities of the materials were examined by the compressive strength of the freeze-dried Alg-HA, Alg-HA-GO, and Alg-HA-GO-F scaffolds. Figure 6B shows the stress and strain curve of the developed composite scaffolds. The compressive strength values of Alg-HA, Alg-HA-GO, and Alg-HA-GO-F are 1.27 MPa, 4.02 MPa, and 3.97 MPa, respectively. The compressive strength of Alg-HA-GO and Alg-HA-GO-F scaffolds was significantly increased by adding 0.1% GO. The increase in the mechanical property was attributed to the reinforcing effect of GO powder contributing to the intermolecular crosslinking in the polymer matrix [30,41]. The achieved mechanical strength is similar to the mechanical properties of human cancellous bone or trabecular bone [8,35,52].

### 2.12. Protein Adsorption Study

The protein adsorption affinity of synthesized scaffolds is essential to study as it affects cell attachment and subsequent proliferation. The addition of graphene oxide and fucoidan shows a greater protein adsorption capacity than the Alg-HA (Figure 6C). In addition, functional groups in fucoidan, such as hydroxyl, carbonyl, oxygen, and sulfates, increase the scaffolds’ protein-adsorbing ability, providing hydrophilicity [38]. Lu et al. and Venkatesan et al. reported that the fucoidan-loaded scaffolds showed better protein adsorption in N, O-carboxymethyl chitosan/fucoidan, and chitosan–alginate–fucoidan, respectively, because of the negatively charged sulfate group in the fucoidan [39,40].

### 2.13. Biomineralization Functionality of the Scaffolds

The principal requirement of the composites in bone regeneration is to achieve mineral formation in the bone-defected site. Such in vivo mineralization can be mimicked in SBF. The capability of the scaffold to develop an hydroxyapatite layer on its surface when submerged in the SBF predicts its potential for osteointegration and its in vivo bone-bonding ability [53,54]. The developed scaffolds incubated with SBF for 30 days reveal apatite deposition on their surface. Furthermore, the SEM images of the biomineralized scaffolds showed the morphology and components represented in Figure 7A. The results show spherical crystal formation on the scaffolds’ surface, which comprises HA particles that act as active spots for crystal precipitation [50]. The qualitative and quantitative elemental composition indicates an increased calcium and phosphate content for Alg-HA-GO and Alg-HA-GO-F composites. In addition, the recorded FT-IR spectra for the mineralized scaffolds completely imitated the peaks of HA, attesting to the mineral formation on the scaffold surface (Figure 7B). The characteristic bands at 590, 632, 991, and 1080 cm^−1^ correspond to the phosphate (PO_4_^3−^) group of the HA phase. At the same time, weak bands at 829 and 1469 cm^−1^ represent the partial substitution carbonate group (CO_3_^2−^) in the HA structure. The literature data clearly show that the IR signals at 1469 and 829 cm^−1^ are presumed for B-type carbonated hydroxyapatite, which is formed by replacing the PO_4_^3−^ groups [55,56]. Bone comprises 4–8% of carbonate content in the mature mineral. Hence, the findings demonstrate that adding GO and fucoidan enhances the osteointegration of the composites, which is suitable for bone regeneration.

### 2.14. In vitro Cell Adhesion, Cell Viability, and Cell Proliferation

Biocompatibility properties emphasize the feasibility of the developed Alg-HA, Alg-HA-GO, and Alg-HA-GO-F composites for bone tissue engineering applications. Hence, cell viability, adhesion, and proliferation on biomaterials are the prominent checks to evaluate the scaffold’s suitability for bone tissue regeneration.

The quantitative analysis using the trypan blue dye exclusion method (Figure 8A) was performed to study the cell proliferation effect of synthesized scaffolds. It is based on the principle that the blue dye stains the dead cells and leaves the live cells unstained. The cells attached to Alg-HA, Alg-HA-GO, and Alg-HA-GO-F after 24 h, 48 h, and 96 h were 0.122 × 10^6^ and 0.150 × 10^6^, 0.145 × 10^6^ and 0.168 × 10^6^, and 0.165 × 10^6^ and 0.197 × 10^6^, respectively, and the count of the live cells after 96 h reached 0.255 × 10^6^ cells per well for the Alg-HA-GO-F composite, which was approximately 1.5 times greater than the control. Concerning the percentage of cell proliferation, by 96 h, Alg-HA-GO-F showed a significant augmentation effect with 50% and 40% when compared to Alg-HA and Alg-HA-GO, respectively [30,38]. Cell proliferation is highly influenced by the composition and structure of the scaffolds [57]. The HA and GO are believed to display surface roughness and beneficial chemical bonding with the host tissue. Thus, it provides the necessary osteoconductive property that favors cell attachment [58,59,60]. While fucoidan loading delivers an enhanced hydrophilicity, abundant functional groups resulted in enriched proliferation [61]. The architectural characteristics of the scaffold’s porosity and pore size play a huge role in cell cultivation [62,63]. The greater the pore size, the more it grows freely. Thus, the results indicate that the Alg-HA-GO-F can provide an environment that is remarkably suitable for murine MSCs proliferation.

The findings in Figure 8B indicate the cell morphology of murine MSCs on Alg-HA, Alg-HA-GO, and Alg-HA-GO-F scaffolds analyzed by SEM. The SEM images of mMSCs after two days of culture demonstrate that the cells adhered to the surface of composites. Furthermore, the appearance of the cells was flatter, with elongations, and well-spread on scaffolds, inferring the excellent interactions between MSCs and the Alg-HA, Alg-HA-GO, and Alg-HA-GO-F scaffold structure. Puvaneshwary et al. reported a mesenchymal stromal cell morphology interaction with the β tricalcium-phosphate–chitosan–fucoidan scaffold [64].

Further, fluorescence staining was performed to examine the viability of the murine MSCs by AO and EB (Figure 8C). All of the composites showed a more significant number of green-stained live cells with few dead cells. Moreover, an increase in the proliferation rate with the addition of each biomaterial to the control was suggested.

### 2.15. Alkaline Phosphatase Activity

During osteogenic differentiation, the alkaline phosphatase enzyme is an early marker expressed by pre-osteoblast cells. Thus, the scaffold’s ability to induce osteogenesis and osteoinduction was examined by ALP activity. The ALP activity of scaffolds at two different time intervals is shown in Figure 9A. The findings demonstrated that the ALP activities of murine MSCs were increased in all three composites, Alg-HA, Alg-HA-GO, and Alg-HA-GO-F, compared to the control. Furthermore, the relative ALP activity of the murine MSCs on Alg-HA-GO-F was greater than Alg-HA and Alg-HA-GO at both time points. Hence, the results unveiled that incorporating fucoidan in the Alg-HA-GO-F boosts osteogenic differentiation in murine MSCs.

Fucoidan is a marine glycosaminoglycan, and its structural similarity enables it to bind with proteins such as cytokines and growth factors. The literature has reported fucoidan’s effect on inducing osteogenic differentiation in osteoblast cells and mesenchymal stem cells. An article recently reported that fucoidan has the highest binding affinity for BMP-2 rather than heparin. The study used the fucoidan complex to deliver BMP-2, resulting in an enhanced ALP and mineralization, confirming its osteo-inductive and osteogenic ability both in vitro and in vivo [65]. Another study used fucoidan with copper sulfide nanoparticles (CuS NPs) to modify chitosan nanofibers. The study results demonstrate the promoted ALP activity in osteoblast cells due to the release of fucoidan and CuS NPs [66]. Similar studies of fucoidan with hydroxyapatite, chitosan, and alginate reported that the fucoidan-incorporated scaffolds were more effective than the other scaffolds [40,67,68].

### 2.16. Calcium Accumulation Study

Calcium accumulation defines late-phase osteogenic differentiation. Therefore, the analysis is essential to understand whether the murine MSCs were stimulated to mature osteoblast cells. VonKossa and alizarin red S experiments used staining to measure the calcium accumulation for 14 and 21 ddays with mMSC. Figure 9C,D display black and red deposits obtained by staining the calcium deposits on the cells. The visual observation indicates that the composites Alg-HA, Alg-HA-GO, and Alg-HA-GO-F and the control exhibited calcium deposition as the cells were cultured in osteogenic differentiation media. However, it was evident that the calcium accumulation was higher at scaffold-loaded fucoidan at both time points. The results were further confirmed by the quantification shown in Figure 9B, similar to the staining images. The calcium formation was higher at 21 days when compared to the 14 days test, as mineralization represents the later stage of osteogenic differentiation.

The surface roughness reports on the cell–substrate interaction uncovers the effect of roughness in determining the cell adhesion, proliferation, differentiation, and the fate of distinct cells. The findings reveal maximum calcium deposition in mMSC’s on Alg-HA-GO-F with a roughness of 259.1 nm, which supports greater cell adhesion and ALP activity. The composites Alg-HA and Alg-HA-GO had a nanoscale roughness of 426.8 nm and 342.9 nm, respectively, on which, the murine MSCs adhesion and differentiation were found to be increased with the decreased nanoscale roughness. From the results, it was important to note that the greater the nanoscale roughness, the lower the stem cell attachment and osteogenesis. Several studies have investigated the impact of topography on MSCs attachment and its osteogenic differentiation. A study investigated the effect of the micro/nanohybrid surface of HA on the stimulation of the osteogenic differentiation of bone marrow MSCs. The results signify that different stimulation mechanisms were followed by micro/nanohybrid structures with micropatterns of 24–28 µm and nanorods of 70–100 nm dimensions [69]. Therefore, along with the osteoconductive and inductive biomaterials, nanoscale roughness plays a key role and contributes to a synergetic effect on the activation of osteogenesis.

However, further research was needed to understand the effect of synthesized scaffolds on gene levels of osteogenesis. One more added limitation is the absence of an in vivo study. An animal experiment could have aided in exploring the potential repair effects and ectopic osteogenesis on bone defects of the fabricated scaffold.

## 3. Materials and Methods

### 3.1. Materials

Sodium alginate was purchased from Sigma-Aldrich, St Albuch, Germany. Nanohydroxyapatite (HA, <200 nm particle size) and nanographene oxide powder (nGO, 15–20 sheets, CAS# 796034-1G) were obtained from Sigma-Aldrich St. Louis, MA, USA. Fucoidan was isolated from *Sargassum ilicifolium* collected from the coast of Tamil Nadu, India, with a degree of sulfation of 0.37 and percentage content of sugar, uronic acid, and protein of 35.6, 2.3, and 6.7%, respectively. Calcium chloride (CaCl_2_) was purchased from HiMedia, Nashik, India. Ethanol was obtained from an analytical CSS reagent Changshu, China. Lysozyme was procured from Sigma Aldrich Co (Darmstadt, Germany). Bovine serum albumin (BSA) was obtained from HiMedia (Mumbai, India). Sodium dodecyl sulfate was purchased from HiMedia (Mumbai, India). The murine multipotent *mesenchymal stem cells* (C3H10T1/2, murine *MSCs*) was purchased from National Centre for Cell Science, Pune, India. Dulbecco’s modified Eagle medium (DMEM) was obtained from HiMedia (Mumbai, India). Fetal bovine serum (FBS), GlutaMAX, and antibiotic ((10,000 U/mL Penicillin-streptomycin) were purchased from Gibco (Grand Island, NE, USA). Ascorbic acid, dexamethasone, and β-glycerol phosphate were obtained from Sigma-Aldrich (St. Louis, MA, USA). Trypan blue dye, acridine orange, ethidium bromide, and magnesium chloride were purchased from HiMedia (Nashik, India). The p-nitrophenyl phosphate was procured from Sigma-Aldrich Co (Darmstadt, Germany). Sodium hydroxide was bought from Merck (Mumbai, India). The staining and fixing reagents alizarin red-S and paraformaldehyde were obtained from Sigma-Aldrich Co (Darmstadt, Germany). The silver nitrate and sodium thiosulfate were obtained from HiMedia (Mumbai, India).

### 3.2. Preparation of Scaffolds

#### 3.2.1. Fabrication of Alginate–Hydroxyapatite (Alg-HA)

The freeze-drying technique was used to perform the fabrication of sodium alginate nano-hydroxyapatite (Alg-HA) [39]. First, where three wt.% of hydroxyapatite was dispersed in milli-Q water, an equal amount of sodium alginate was added under continuous stirring. Next, the solution was further allowed to stir for 1 h until the formation of a white-colored, thick gel solution was observed. Later, the solution was cast in different tissue culture plates to achieve a certain thickness and diameter of scaffolds. Subsequently, the culture plates were frozen overnight at −20 °C, followed by lyophilization. Then, the resulting lyophilized scaffolds were crosslinked or immersed with 10% CaCl_2_ for 10 min. Next, the crosslinked scaffolds were soaked in ethanol solution for 10 min, followed by a vigorous water wash. Later, the scaffolds were frozen and subjected to lyophilization.

#### 3.2.2. Preparation of Alginate–Hydroxyapatite–Graphene-Oxide (Alg-HA-GO) Scaffolds

GO (0.1%) was added to the 3% *w*/*w* homogenous mixture of nanohydroxyapatite and sodium alginate. Next, the solution was cast in various tissue culture plates, followed by crosslinking and freeze-drying processes similar to Alg-HA composite preparation.

#### 3.2.3. Fabrication of Fucoidan-Loaded Alginate–Hydroxyapatite–Graphene-Oxide (Alg-HA-GO-F)

The 0.1 wt.% of purified fucoidan from *Sargassum ilicifolium* solution was prepared in milli-Q water. Then, the fucoidan sample was loaded dropwise to the solution containing nanohydroxyapatite, sodium alginate, and nano graphene oxide. Further steps were similar to the Alg-HA mentioned above for composite preparation.

### 3.3. Chemical Characterization of the Developed Composite Scaffolds

The chemical bonds in the scaffolds composite Alg-HA, Alg-HA-GO, and Alg-HA-GO-F were investigated with FT-IR spectroscopy (Shimadzu Corp Tokyo, Japan). The identification of crystalline phases of the material present in the scaffolds was monitored using the Malvern PANalytical Empyrean 3rd Gen X-ray diffractometer. The CuKα radiation (λ = 0.154 nm, 45 kV, 30 mA) was used as a source, and the angle scans were examined in the diffraction range of 4–80° (2θ), with step size 0.02°. Thermogravimetric analysis was carried out using TGA 4000, SPECTRUM-2 (Perkin Elmer, Singapore). The samples were kept in the presence of nitrogen gas running at 30.0 mL/min. Later, the specimen was heated from 37 °C to 600 °C at 10 °C/min. Subsequently, the obtained TGA graph of each sample signifies the percentage weight loss against the temperature. After fabrication, the scaffolds’ surface analysis was performed through an optical microscope (Magnus, MLXi plus). The morphology of the scaffolds was examined using FESEM (GEMINI 300, Carl Zeiss, Jena, Germany). Samples were gold-sputtered to minimize sample charging. The scaffolds were imaged at 15 kV of beam energy. Next, the scaffolds’ elemental composition was examined by energy-dispersive X-ray spectroscopy (EDS—EDAX AMETEK). The surface roughness of Alg-HA, Alg-HA-GO, and Alg-HA-GO-F was inspected using AFM (Scanning Tri-AFM, APR Research, Basovizza, Italy). Topographic images were captured, and the Ra values were obtained from different scan areas of each sample.

### 3.4. Porosity and Pore Size

The porosity is the void space in the solid, independent of material property. The total porosity was measured using the liquid displacement method [47]. In brief, the freeze-dried scaffolds of equal quantity were immersed in a specific volume of ethanol (V_1_). The increased volume was measured as V_2_. After 48 h, the scaffolds with some volumes of absorbed ethanol (Vs) were taken out, and the remaining volume of ethanol was marked as V_3_. The increased volume of the scaffolds was evaluated using Equation (1).
(1)$$Vs=V_{2}-V_{3}$$

The porosity percentage was calculated by following Equation (2).
(2)$$\%\;Porosity=\frac{V_{1}-V_{3}}{Vs}\times 100$$

The test was conducted with five specimens, and the mean porosity and standard deviation were measured. Later, the pore sizes of Alg-HA, Alg-HA-GO, and Alg-HA-GO-F scaffolds were measured by analyzing the obtained FESEM images using ImageJ software.

### 3.5. Swelling/Retention Measurement

Each scaffold’s known weight (W_i_) was immersed in phosphate buffer saline (PBS) for different time intervals (2, 4, 6, 8, 10, and 24 h). After each time point, the swollen scaffolds were taken out, soaked on tissue paper, and weighed. The weight gained was noted as W_s_. The experiment was conducted until there was no weight gain observed. Subsequently, the retention ability of the scaffold was measured by placing the swollen scaffolds inside a tube that contained filter paper at the bottom. The container was centrifuged at 500 rpm for 3 min to remove the solvent. Further, the scaffold was immediately weighed and marked as (W_d_). Finally, the swelling and retention rate of the scaffold was calculated using the following Equations (3) and (4).
(3)$$\%\;Swelling=\frac{W_{s}-W_{i}}{W_{i}}\times 100$$
(4)$$\%\;Retention=\frac{W_{d}-W_{i}}{W_{i}}\times 100$$

### 3.6. In Vitro Biodegradation

The biodegradation of the scaffolds was studied. Briefly, the initial weight of the scaffolds was noted as W_o_. Next, the equally weighed scaffolds were immersed in PBS solution containing lysozyme and incubated at 37 ℃ at different time intervals (15 and 30 days). After incubation, the scaffolds were removed from the PBS with the lysozyme solution and freeze-dried. The weight of the scaffolds after degradation is marked as Wt. The weight loss percentage is evaluated using Equation (5).
(5)$$\%\;Weight\;loss=\frac{W_{o}-W_{t}}{W_{o}}\times 100$$

### 3.7. Fucoidan Loading/Release Study

Loading capacity and encapsulation efficiency of the fucoidan in the scaffolds were examined using an ultraviolet spectrophotometer (Shimadzu UV-1800, Kyoto, Japan) [70]. For this, a total amount of fucoidan (X mg) was used while synthesizing the scaffolds. The fucoidan content was released after centrifuging the synthesized scaffold in water at 5000 rpm for 3 min. As per the calibration curve, the released fucoidan was quantified spectrophotometrically at λmax = 260 nm. In addition, the percentage loading capacity (% LC) was measured using Equation (6).
(6)$$\%\;LC=\frac{Total\;amount\;of\;fucoidan-free\;amount\;of\;fucoidan}{Total\;amount\;of\;fucoidan\;}\times 100$$

A freeze-dried fucoidan-loaded scaffold was soaked in a dialysis bag holding 1X PBS submerged in a 30 mL glass bottle with milli-Q water as outer media. Another set with Alg-HA-GO-F composite was immersed in 10 mL SBF (pH-7.25) to investigate the fucoidan release profile. Both the bottles were continuously shaken at 100 rpm at 37 °C, and, at definite time intervals, 1 mL aliquot of water and SBF was withdrawn and immediately replaced with an equal volume of respective solution. The fucoidan release was measured using a UV spectrophotometer at 260 nm according to the calibration curve. Further, the percentage release was calculated according to Equation (7).
(7)$$\%\;Release=\frac{Concentration\;of\;fucoidan\;(µg/mL)\times dilution\;factor}{Total\;amount\;of\;fucoidan\;loaded\;in\;the\;scaffold}$$

### 3.8. Mechanical Properties

A compressive test was performed using a universal testing machine to assess the mechanical properties of freeze-dried Alg-HA, Alg-HA-GO, and Alg-HA-GO-F scaffolds (TECSOL, India). The scaffolds with 12 mm diameter and 9 mm height were used. The stress–strain curve was plotted by measuring the compressive strength of scaffolds with Equation (8), where F is the applied force and A is the sample area.
σ = F/A(8)

### 3.9. Protein Adsorption Study

The protein adsorption ability of scaffolds was measured according to the previous study [40]. The scaffolds were immersed in 0.5 mL FBS and incubated at 37 °C at different time points (1.5, 3, and 5 h). After incubation, the scaffolds were washed with PBS and a 1% sodium dodecyl sulfate solution was added. Later, the solution was centrifuged with the scaffolds at 4 °C, and the collected supernatant was used for protein estimation by UV spectrophotometric assay using BSA standard calibration curve [R^2^ = 0.9997].

### 3.10. In Vitro Biomineralization

The biomineralization effect of developed scaffolds was performed using simulated body fluid (SBF). First, the SBF solution was prepared as described by Kokubo et al. [71]. Next, the fabricated scaffolds were immersed in the SBF solvent for 30 days at 37 °C. After the incubation period, scaffolds were washed with milli-Q and air-dried. Further, SEM-EDAX and FT-IR were employed to observe the formation of the mineral on the sample’s surface and its elemental composition.

### 3.11. In Vitro Cell Attachment, Viability, and Proliferation Studies

Fluorescent staining was performed to measure the cell viability using green dye (AO—acridine orange) and red dye (EB—ethidium bromide). The cells were seeded in the 24-well plates at a density of 0.03 × 10^6^ cells/well. The next day, the cells were treated with flakes of fabricated scaffolds. After 2 days, the media were aspirated and cells were given PBS wash, followed by the addition of chilled methanol. The plate was kept aside at room temperature for 30 min. Next, cells were stained with each dye for 15 min at 37 °C in the dark, where AO-EB stained the live and dead cells. Subsequently, fluorescence images of cells were captured using a fluorescence imager (ZOE, Bio-Rad, Hercules, CA, USA) [70].

To study the cell proliferative effect, cells at a density of 0.03 × 10^6^ cells/well were seeded in the 24-well plates and exposed to the developed scaffold. After 24 h and 48 h, media were removed, and the cells were trypsinized in order to detach. Then, the collected cells were suspended with an equal volume of media. Further, the cells were counted using a haemo-cytometer by mixing the cell suspension with 0.4% trypan blue dye in a 1:1 ratio.

Scanning electron microscopy assessed the cell adhesion and morphology studies of the cell scaffold. Initially, composites were immersed in the media overnight, followed by cell seeding. After two days of culture, the cell-distributed scaffolds were given PBS wash and fixed using formaldehyde. After, dehydration with ethanol at increasing concentrations (50, 70, 96, and 99.9%) for 3 min was repeated three times in each solution. Finally, the cell–scaffold constructs were air-dried overnight before the assessment under SEM (Carl Zeiss, EVO 10, Jena, Germany) [72].

### 3.12. Alkaline Phosphatase Activity Assay

The cell density of 0.03 × 10^6^ cells/well was cultured in 24-well plates. The confluent cells were treated with composite scaffold Alg-HA, Alg-HA-GO, and Alg-HA-GO-F for 7 and 14 days in an osteogenic differentiation medium. The media were changed every two days. After incubation, the cells were washed with 1X PBS twice, followed by homogenization of the cells using 100 µL 25 mM carbonate buffer containing 0.2% Triton X-100 (pH-10.3). Further, 50 µL 250 mM carbonate buffer containing 2.5 mM MgCl_2_ and 15 mM p-nitrophenyl phosphate was added. Soon after, the plate was incubated at 37 °C for 30 min, and 1 M NaOH was used to stop the reaction. The absorbance of the reaction was measured at 405 nm. The cells cultured in osteogenic differentiation media were taken as control.

### 3.13. Calcium Accumulation Study

Alizarin red-s and Von Kossa staining scrutinized the calcium deposition. The murine MSCs at a density of 0.03 × 10^6^ cells/well were seeded in a 24-well plate and cultured with scaffolds for 14 and 21 days. After 14 and 21 days, cells were fixed with 4% paraformaldehyde for 30 min and stained with 40 mM alizarin red s solution. After 30 min of incubation, the calcium-stained cells were imaged using a microscope (Primovert, Carl Zeis, Jena, Germany). Further, 10% cetylpyridium chloride in 10 mM sodium phosphate solubilized the cells for quantification. After incubation at 37 °C for one hour, the absorbance was measured at 570 nm.

In addition, Von Kossa staining measured the mineralization. After 14 days of culture, cells were fixed using 4% paraformaldehyde, and 1% silver nitrate solution was added. Subsequently, the plate was exposed to light for 30 min, and the cells were rinsed with distilled water after adding 5% sodium thiosulfate for two minutes. After alcohol wash, the formation of black deposits was observed [73].

### 3.14. Statistical Analysis

All of the data are expressed as means ± the standard deviation of a minimum of three replicates for each composite in every experiment, using Origin Lab (OriginLab, MA, USA) and GraphPad Prism 8.0. (GraphPad Software, San Diego, CA, USA)

## 4. Conclusions

We developed fucoidan-loaded Alg-HA-GO osteoinductive scaffolds for bone tissue repair and regeneration. The FT-IR confirms the ionic interaction between the materials, the XRD shows the sharp crystalline peaks of HA, and the other materials used are amorphous. The surface properties analyzed by FESEM and AFM reveal that the pore size between 96–460 μm and has 426–260 nm nanoscale roughness from all of the composites. The slower rate of biodegradation and increased protein adsorption of Alg-HA-GO-F compared to Alg-HA and Alg-HA-GO are suitable for bone regeneration. The slow and sustained release of fucoidan positively influences osteogenic differentiation. The composites can form biomineralization using SBF. Porous surfaces with nanoscale topography stimulate the cell attachment, proliferation, and differentiation of stem cells. The in vitro studies of scaffolds showed an enhanced murine MSCs proliferation, alkaline phosphatase activity, and calcium deposition. The fucoidan containing alginate, hydroxyapatite, and graphene oxide will be excellent biomaterial for orthopedic applications.

## Figures and Tables

**Figure 1 marinedrugs-20-00589-f001:**
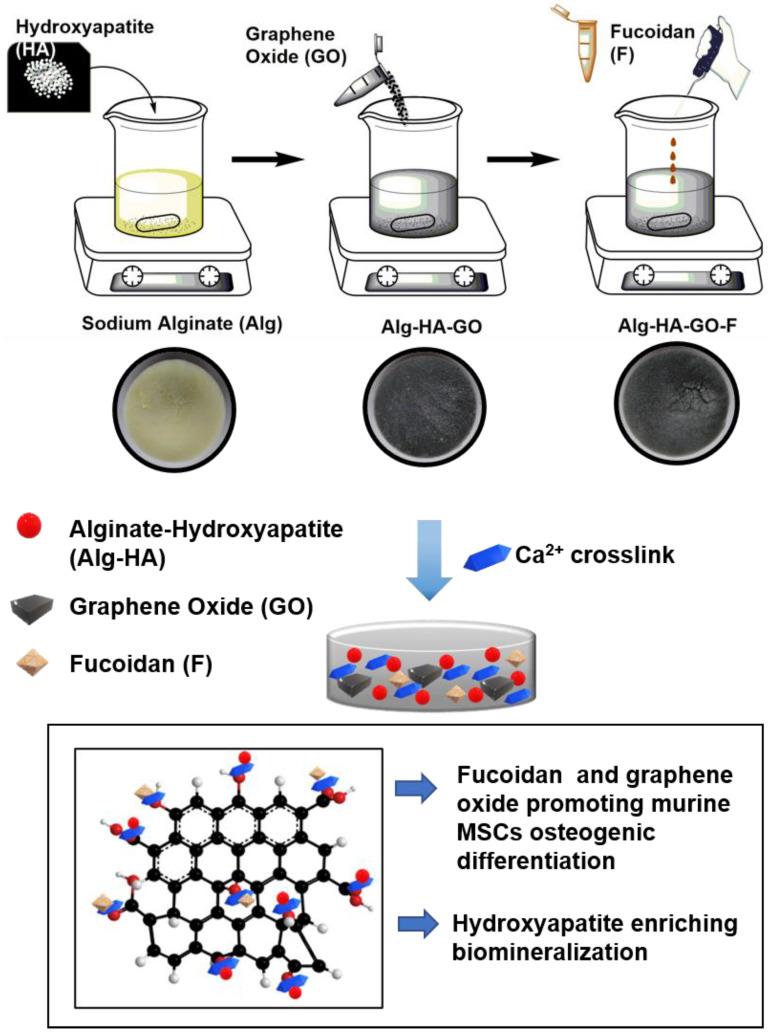
The schematic representation of Alg-HA-GA-F scaffolds preparation of the procedure.

**Figure 2 marinedrugs-20-00589-f002:**
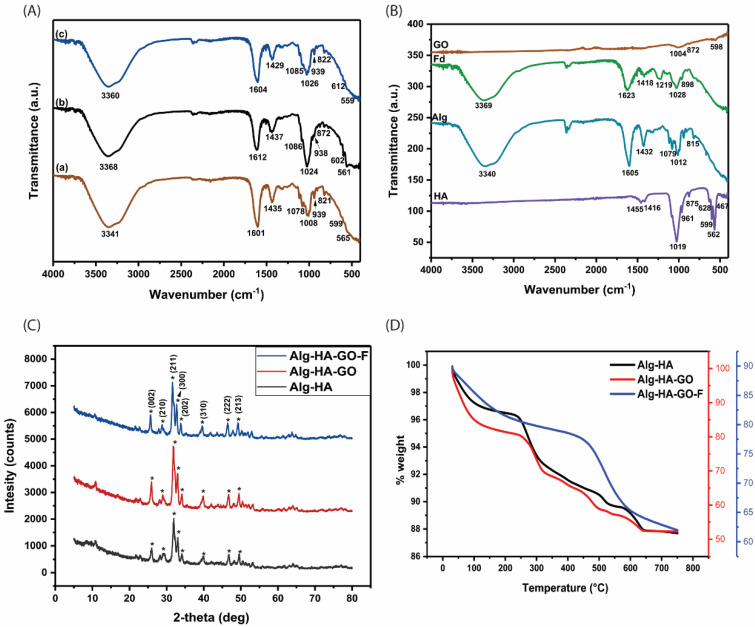
Fourier transform infrared spectra of A. combinations where (**A**) Alg-HA, (**B**) Alg-HA-GO, (**C**) Alg-HA-GO-F scaffolds, B. FT-IR spectra of materials, hydroxyapatite (HA), sodium alginate (Alg), graphene oxide (GO), and fucoidan (Fd), C. X-ray diffraction spectra, and (**D**). thermogravimetric analysis for the developed Alg-HA, Alg-HA-GO, and Alg-HA-GO-F scaffolds.

**Figure 3 marinedrugs-20-00589-f003:**
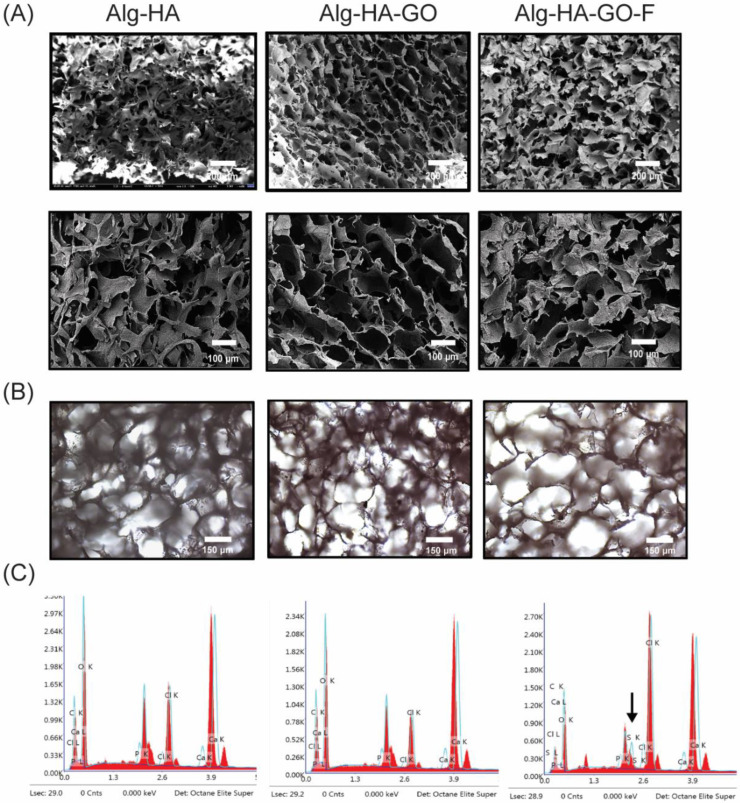
(**A**) FESEM micrographs at 100X and 200X magnification, (**B**) optical microscopy images at 20X, and (**C**) EDAX analysis of the developed scaffolds, Alg-HA, Alg-HA-GO, and Alg-HA-GO-F.

**Figure 4 marinedrugs-20-00589-f004:**
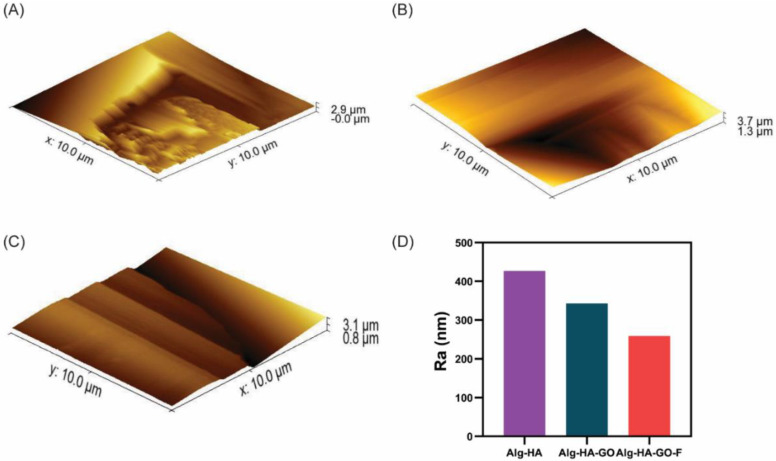
AFM 3D images of surface roughness analysis for (**A**). Alg-HA, (**B**). Alg-HA-GO, and (**C**). Alg-HA-GO-F. (**D**). Graph of the average roughness (R_a_) at different positions using AFM images.

**Figure 5 marinedrugs-20-00589-f005:**
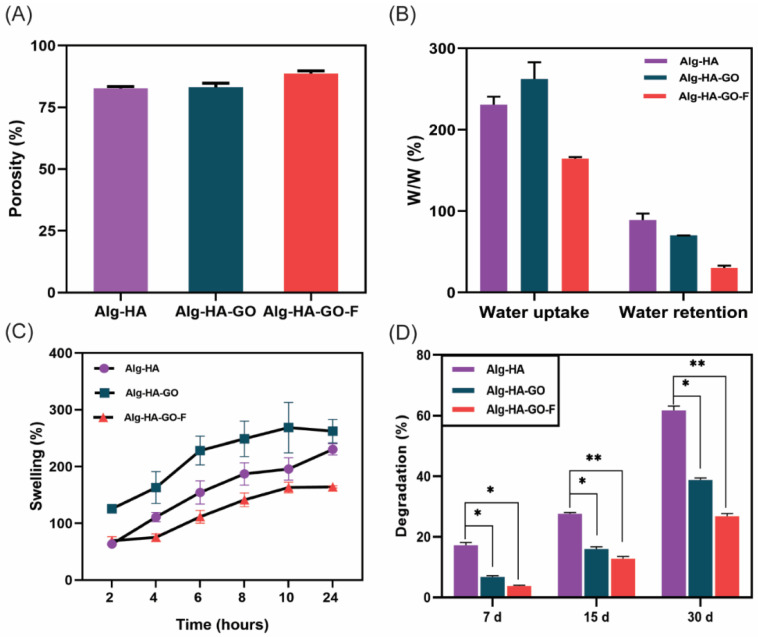
Physiochemical characterization of as-synthesized Alg-HA, Alg-HA-GO, and Alg-HA-GO-F scaffolds: (**A**) porosity measurement by solvent displacement method, (**B**) water uptake and retention study after 24 h, (**C**) swelling study at different time points (2–24 h), and (**D**) in vitro biodegradation in PBS containing lysozyme. Error bars represent standard deviation. Here, data with * *p* < 0.05 and ** *p* < 0.01 were compared with the Alg-HA.

**Figure 6 marinedrugs-20-00589-f006:**
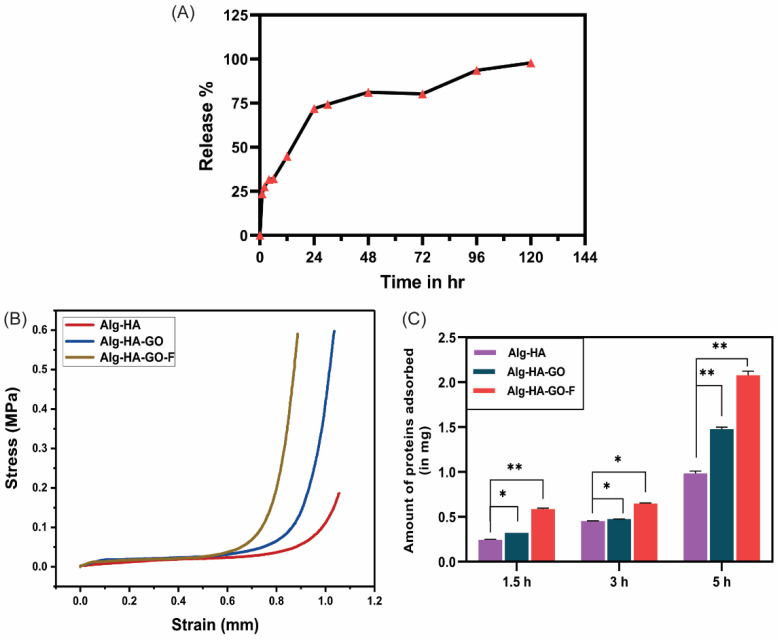
In vitro release study of fucoidan from the scaffolds Alg-HA-GO-F using (**A**). PBS, (**B**). compression stress–strain curves of Alg-HA, Alg-HA-GO, and Alg-HA-GO-F scaffolds under compression, and (**C**). protein adsorption study of the developed Alg-HA, Alg-HA-GO, and Alg-HA-GO-F composite scaffolds. Error bars represent standard deviation. Data with * *p* < 0.05 and ** *p* < 0.01 compared with the Alg-HA.

**Figure 7 marinedrugs-20-00589-f007:**
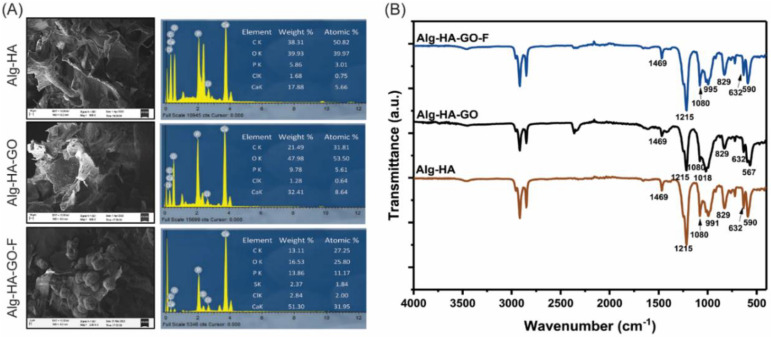
Apatite formation ability of Alg-HA, Alg-HA-GO, and Alg-HA-GO-F scaffolds: (**A**). SEM-EDAX analysis and (**B**). FT-IR analysis for in vitro biomineralization studies in SBF for 30 days.

**Figure 8 marinedrugs-20-00589-f008:**
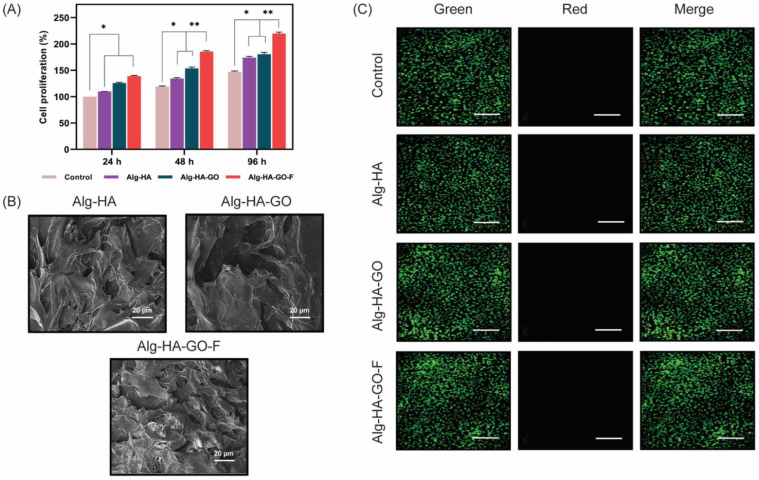
(**A**). The cell proliferation effects of Alg-HA, Alg-HA-GO, and Alg-HA-GO-F composites at 24 and 48 h. Then, cells were harvested to calculate the percentage of cell proliferation using a trypan blue assay. Error bars and columns represent standard deviation and means. For example, here, * at *p* < 0.05 and ** at *p* < 0.01 indicate statistical difference, (**B**). SEM images showing the murine MSCs attachment on 3D scaffold surface after 3 days of culture, and (**C**). AO/EB staining for cell–scaffold construct after 2 days of culture, scale bar = 100 μm.

**Figure 9 marinedrugs-20-00589-f009:**
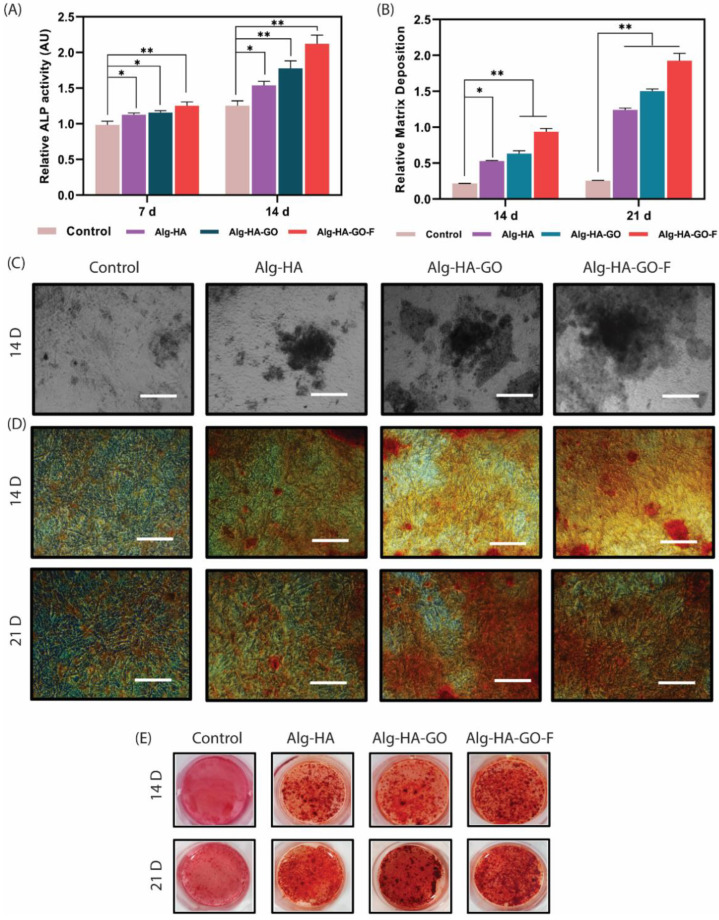
Effects of developed scaffolds on (**A**). ALP activity for 7 and 14 days (*n* = 3). Cells were treated with composites containing Alg-HA, Alg-HA-GO, and Alg-HA-GO-F, and the generated p-nitrophenol detected ALP activities. Error bars and columns represent standard deviation and means. Here, * at *p* < 0.05 and ** at *p* < 0.01 indicate statistical difference. (**B**). Optical density measured at 570 nm when solubilized with cetylpyridinium chloride solution. Scale bar = 150 μm. All data represent the mean ± s.d. of three independent experiments. * *p* < 0.05 and ** *p* < 0.01 compared with the control cells. Determination of calcium accumulation using: (**C**). Von Kossa staining after 14 days of murine MSCs culture, Scale bar = 150 μm (**D**,**E**). Alizarin Red S, murine MSCs cultured for 14 and 21 days.

## Data Availability

Not applicable.

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
