# Peer review of "Fucoidan-Incorporated Composite Scaffold Stimulates Osteogenic Differentiation of Mesenchymal Stem Cells for Bone Tissue Engineering"

_marinedrugs, 2022, doi:10.3390/md20100589_

Round 1

Reviewer 1 Report

The authors developed a calcium crosslinked sodium alginate-nano hydroxyapatite-nano graphene oxide (Alg-HA-GO-F) encapsulating fucoidan and control it release to stimulate osteogenic differentiation of mesenchymal stem cells for bone tissue regeneration. This study was well organized and very interesting. I recommend publishing with minor revisions. My comments are shown below.

1. The authors need to provide some recently published papers (https://doi.org/10.1002/smll.202200416; https://doi.org/10.1016/j.carbpol.2022.119286; https://doi.org/10.1016/j.carbpol.2021.119035; https://doi.org/10.1002/jbm.a.37334) related to fucoidan-incorporated materials for bone tissue engineering to highlight the importance of this study.

2. Basic information on fucoidan such as molecular weight, fucose content and degree of substitution of sulfate groups must be provided.

3. The scale bars of the composite scaffolds in the SEM image (Figure 3A and Figure 8B) are not clear. Please relabel the images with new scale bars.

4. The scale bar of the cell images in Figure 8A is omitted.   

5. The authors need to provide the DTG curve along with the TGA curve to make it easier to read the transition temperature.

6. I suggest that the authors combine the release profiles of fucoidan in PBS and SBF in one graph to make it easier to compare the release properties in different dissolution medium.

7. Please quantify calcium deposition in Alizarin Red stained MSCs.

Author Response

We have enclosed the response to the reviewer's comments in the attachment

Reviewer 2 Report

1.     The scaffolds composed of alginate, hydroxyapatite and graphene have been reported in previous works for the culture of osteoblasts (doi.org/10.1016/j.matchar.2015.07.016), differentiation of stem cells (doi.org/10.1007/s10856-020-06467-6), bone regeneration and so on. The specific contribution of this research is not clear enough.

2.     According to Figure 3, the porous structure is lamellar with large pore sizes. Does the structure result in poor and fragile mechanical strength? Is the mechanical properties suitable for bone tissue engineering? What is the duration of scaffold structure in the cell culture process?

3.     What is the scale bar in Figure 3 (B)? Is it consistent with the pore sizes observed in Figure 3 (A)?

4.     It was mentioned that fucoidan crosslinked among alginate, HA and GO, and the degradation was thus decreased. Is there any clear indicator in FTIR spectra, thermal analysis or mechanical strength caused by the crosslinking reaction?

5.     The addition of fucoidan resulted in a lower swelling ratio. Please explain the mechanism for this.

6.     According to Figure 6, almost all the fucoidan would be released from scaffolds after 120 hours. If there is a significant interaction between fucoidan and the other component, is it reasonable for such a fast and completed release?

7.     In the experiments of cell proliferation, what is the thickness of scaffolds? Did cells proliferate into the scaffolds?

8.     What is “The confluent cells were treated with composite scaffold” in Line 581? Does it mean that the cells were just in contact with cells? If yes, what were the scaffold size, medium amount, and cell number in this experiment? Is the cell proliferation and differentiation different if fucoidan is added into the culture medium instead of being released from scaffolds?

9.     In Figure 9 (C), (D) and (E), the differences of biomineralization were not clear enough.

Author Response

We have enclosed the response to the reviewer's comments in the attachment.

Reviewer 3 Report

The manuscript submitted to Marine Drugs entitled ‘Fucoidan encapsulated composite scaffold stimulates osteogenic differentiation of mesenchymal stem cells for bone tissue regeneration describes the physiochemical characterization of different scaffolds for bone graft substitutes for orthopedic applications. It is an interesting work with the potential to be published in Marine Drugs, however, it needs to be improved and some points need to be clarified.

Major revisions

- The introduction needs to be improved. Especially, the text between lines 59-73, because it is like a review article text and is not necessary for this context.

- Authors should add a discussion section. In the results section, some results are discussed, but other results are just presented without appropriate discussion. So, an integrated discussion of the results will improve the study and allow the comparison with other materials, mainly, showing the advantages of the materials developed in this study.

- What do the authors mean by fucoidan encapsulated in the composite material? Because by the scheme in figure 1, fucoidan is added to a matrix that is already gelled. Is the fucoidan adsorbed on the matrix or is it encapsulated? This aspect needs to be clarified in the manuscript.

- In fluorescent staining to measure the cell viability the authors said ‘After 2 days of cell culture with scaffolds, the cells were stained with each dye for 15 min at 37 °C in the dark, where AO - EB stained the live and dead cells.’ In the images of figure 8C red cells are not observed. This assay should have a control to confirm that dead cells were being detected. Another question that arises is: did the authors collect the cells that were in the supernatant (the cells that did not adhere and that would be dead) or only the cells that adhered to the materials were analyzed in this assay?

- Was any control performed with cells in a medium without osteogenic inducers? Because in my experience, analyzing the images in Figure 9, the mesenchymal stem cells in osteogenic medium, after 21 days, should be more differentiated, and have a more evident oil red stain. It would be good to compare with cells in a non-osteogenic medium.

Minor revisions

- Authors should use the abbreviation for the seaweed species (S. ilicifolium) after first mentioning it in full (Sargassum ilicifolium). Authors also should uniformize FTIR or FT-IR spectroscopy.

- How Fucoidan was obtained?

- What do the authors mean by fucoidan encapsulated in the composite material?

- Lines 86-87: ‘Several studies have proven the spectra of bioactivities of fucoidan, including antioxidant, anti-inflammatory, antiviral, anti-tumor, and its role in bone repair [19]. Reference 19 is a review article. Authors should avoid using a review article to refer to activities described for fucoidan. Studies showing these results should be referenced.

- In line 343 ‘The cells attached to Alg-HA-GO-F after 24 h and 48 h were much more than the control,’ the expression much more must be replaced by the quantitative value obtained. Another aspect is that a 15-20% increase or decrease in cell proliferation may be statistically significant, but in biological terms, it is not relevant.

- How was the number of adhered cells determined? Because a 24 and 48 h assay does not allow to determine of cell adhesion, since proliferation has already occurred. This should be corrected in line 343.

- Figure 9 needs to be corrected (Alg-GO by Alg-HA).

- In the methodology, some aspects should be better detailed to clarify some points. For example, in the In vitro cell adhesion, cell viability, and cell proliferation tests, cells were plated on the culture plate and the material was dissolved in the medium and added after the adhered cells, or along with the cells, or the material was placed at coat the plate well and the plated cells on top of the material? What was the control used?

-Why was ALP activity assessed at 7 and 14 days? ALP is an early marker that has a peak of activity before 7 days. This peak is normally seen earlier when cells are exposed to hydroxyapatite materials.

- The authors observed an increase in cell proliferation in materials with GO and Fucoidan and describe that ALP activity and extracellular matrix production also increased in these matrices. It would be interesting to present the results of ALP activity and extracellular matrix production, normalized, for example, by the amount of DNA or total protein. Because, normally, when cells start the osteogenic differentiation, cell proliferation slows down. So, a greater amount of cells does not necessarily mean a greater % of differentiated cells.

Author Response

(The authors gave the same response as above.)

Reviewer 4 Report

It is a nicely written manuscript describing the potential application of fucoidan encapsulated Alg-HA-GO scaffold for bone tissue engineering. The methodology is sound and the conclusion is supported by the findings. Just one comment, the authors should discuss the limitations and future prospects of this study. 

Author Response

(The authors gave the same response as above.)

Round 2

Reviewer 2 Report

1. The murine MSCs in SEM images were not clear. The morphologies and lamellipodia/filopodia were not recognizable. The fixed cells should be dried by using a freeze dryer or super-critical dryer instead of air drying.

2. The addition of fucoidan decreased the swelling of scaffolds. According to the authors’ responses, it was caused by the crosslinking interaction between materials and calcium chloride, resulting in a low swelling ratio. Since Alg-HA, Alg-HA-GO and Alg-HA-GO-F were all treated with 10 % CaCl2 solution, why was the crosslinking density of Alg-HA-GO-F higher than the others? The relative explanation should be added into the revised manuscript.

3. In the authors’ responses, it was mentioned that “SBF formation of appetite restricts the release and leads to a prolonged release profile.” However, what was the controlled group for this prolonged release? In Fig. 6 (A), there was only one releasing curve of Alg-HA-GO-F without any controlled group.

4. In comment 7 of the cover letter, the authors did not answer the question that “Did cells proliferate into the scaffolds?” What were the flakes used in cell proliferation? Were there still porous structures in flakes? If the flakes were highly porous, were they transparent enough for the AO/EB staining? The details in the experimental procedures should be clarified in the revised manuscript.

Reviewer 3 Report

The manuscript submitted to Marine Drugs entitled ‘Fucoidan encapsulated composite scaffold stimulates osteogenic differentiation of mesenchymal stem cells for bone tissue regeneration’ has been revised and although the manuscript continues without the 'Results and Discussion' section or just a 'Discussion' section, the discussion of the results was improved. In general, the authors answered the questions, clarifying the doubts about the methodology and results. 

Author Response

We thank you for your valuable suggestions and critical review that supported the improvisation of the manuscript.